# Technical Evaluation of a New Medical Device Based on Rigenase in the Treatment of Chronic Skin Lesions

**DOI:** 10.3390/bioengineering10091022

**Published:** 2023-08-29

**Authors:** Eugenia Romano, Claudio Campagnuolo, Roberta Palladino, Giulia Schiavo, Barbara Maglione, Cristina Luceri, Natascia Mennini

**Affiliations:** 1Farmaceutici Damor S.p.A., Via E. Scaglione 27, 80125 Napoli, Italy; eugenia.romano@farmadamor.it (E.R.); claudio.campagnuolo@farmadamor.it (C.C.); roberta.palladino@farmadamor.it (R.P.); giulia.schiavo@farmadamor.it (G.S.); 2Dipartimento di Chimica “Ugo Schiff”, Università degli Studi di Firenze, 50019 Sesto Fiorentino, Italy; cristina.luceri@unifi.it (C.L.); natascia.mennini@unifi.it (N.M.)

**Keywords:** wound dressing, chronic wound healing, skin injuries, *Triticum vulgare* aqueous extract, Rigenase

## Abstract

Chronic wound is characterized by slow healing time, persistence, and abnormal healing progress. Therefore, serious complications can lead at worst to the tissue removal. In this scenario, there is an urgent need for an ideal dressing capable of high absorbency, moisture retention and antimicrobial properties. Herein we investigate the technical properties of a novel advanced non-woven triple layer gauze imbibed with a cream containing Rigenase, an aqueous extract of *Triticum vulgare* used for the treatment of skin injuries. To assess the applicability of this system we analyzed the dressing properties by wettability, dehydration, absorbency, Water Vapor Transmission Rate (WVTR), lateral diffusion and microbiological tests. The dressing showed an exudate absorption up to 50%. It created a most environment allowing a proper gaseous exchange as attested by the WVTR and a controlled dehydration rate. The results candidate the new dressing as an ideal medical device for the treatment of the chronic wound repairing process. It acts as a mechanical barrier providing a good management of the bacterial load and proper absorption of abundant wound exudate. Finally, its vertical transmission minimizes horizontal diffusion and side effects on perilesional skin as maceration and bacterial infection.

## 1. Introduction

The skin is the largest sensor organ of the body and constitutes about the 15% of its total weight [1]. It provides a protective barrier against mechanical, thermal, and physical injuries, pathogens, and chemical agents [2]. It also plays a key role in thermoregulation, water loss prevention, vitamin D3 synthesis and endocrine, exocrine, and immunological activity [3]. These protective and regulative functions could be affected by several internal and external factors including cutaneous damages [e.g., cuts, surgical incisions, metabolic disorders, ulcers] [4].

These cutaneous damages can be classified as acute or chronic according to the duration and nature of the wound-healing process [5]. Acute wound mainly results in traumatic [i.e., incisions, abrasions, puncture, avulsions, burns, and lacerations] and postoperative conditions [6,7]. In these cases, the skin’s structural and functional integrity is generally restored within 12 weeks through the activation of specific biochemical pathways (as the cytokines and growth factors cascade) [2]. On the contrary, a chronic wound is a complex environment characterized by a persistent inflammatory and proliferative stage [8]. This pathological condition recognized as hard-to-heal wound [9], promotes the accumulation of metalloproteinases, collagenases, and elastases, which prematurely degrade collagen and growth factors [10]. Furthermore, the low oxygen tension generates a hypoxic environment leading to fibroblast proliferation and, consequently, to increased tissue fibrosis [11]. In addition, as a warm and moist area, the chronic wound may give a large opportunity for microbial growth holding up the healing process [12]. Microbial infections are univocally recognized as key factors affecting the healing process [13,14]. Indeed, microorganisms tend to interact with chronic wounds at four different levels: contamination, colonization, critical colonization, and infection [15].

As stated in the TIMERS protocol [9], the wound repair and regeneration are likely to occur in hard-to-heal wounds once the wound bed has been adequately prepared [9]. Then an appropriate dressing has to be chosen according to physical, chemical, and biological properties for the treatment of the specific type of wound [16].

Currently, technological progress in the field of chronic wound treatment has rapidly increased. Several types of wound dressings are clinically available. Even though there isn’t a univocal consensus about the dressing characteristics, some properties are crucial for healing progress. An ideal wound dressing should be biocompatible, non-toxic, non-allergenic, conform to the wound surface, and easy to remove without damage to the skin. To recreate the optimal condition for tissue regeneration, it is necessary to maintain a moist environment with the proper temperature, promote a gaseous exchange and prevent bacterial infection. *S. aureus* is considered the most problematic bacterium in traumatic, surgical, and burn infections [17]. Moreover, *S. aureus* and *K. pneumoniae* are known as the principal responsible for biofilm formation, a major virulence factor contributing to the chronicity of infections [18]. Thus, the removal of such a highly associated risk factor as the microbial infection identified during the comprehensive wound care assessment (TIMERS elements), supports the wound healing.

Furthermore, wounds with high exudate, as ulcers and granulating wounds, require a high absorption capacity and rapid dehydration to avoid maceration [19,20].

The traditional wound dressings are mostly represented by gauzes and bandages. They usually comprise cellulose, cotton fibers, viscose, and polyester materials with the great advantages of being highly hygroscopic and biocompatible for the skin [16,21]. Nonetheless, some of the main limitations are the high evaporation, low absorbency, and fiber loss. Indeed, due to their fibrous nature, cotton bandages could stick to damaged surfaces during wound cleaning and debridement as well as gauzes during their removal [22]. On the contrary, non-woven materials have the advantages of being non-occlusive systems with a high absorbency and reduced healing time of wounds compared with traditional dressings [23,24]. So, they result to be more appropriate for the treatment of chronic wounds.

Herein, a novel non-woven wound dressing registered under the brand name of Fitostimoline Plus Garze ADVANCE (FPA) was tested. It is a sterile medical device (MD IIB) for dermatological use consisting of a single dose Polyethylene Terephthalate (PET) non-woven gauze, impregnated with hydro-dispersible cream. The rationale of the study was to evaluate the exudate interaction capability of a composite matrix fabric assembled in a triple layer system and soaked with a hydro-dispersible cream based on Rigenase. It is an aqueous extract of *Triticum vulgare*, obtained from the whole germinated plant. This vegetable matrix compound is involved tissue-repairing activity due to its antioxidant capacity and a moisturizing action [25,26].

Our results showed the ability of this advanced system to act as a protective barrier against the external environment and to keep the proper conditions for a rapid and correct re-epithelialization of the injured skin in hard-to-heal chronic wounds.

## 2. Materials and Methods

### 2.1. Fitostimoline Plus Garze ADVANCE

Fitostimoline^®^ Plus Garze ADVANCE is a sterile Medical Device (MD) for dermatological use consisting of single dose PET non-woven triple layer gauze, impregnated with hydro-dispersible cream. The external Monofilament Fabric and the internal Monofilament Knitted Fabric were provided by Sefar Group (Freibach, Switzerland). Each device contains Rigenase^®^, 20% water solution of Polyhexamethylene biguanide (PHMB), Glycerin, Macrogols 400, 600, 1500, 4000, Purified water. Terminal sterilization by Gamma radiation according to EN ISO 11137-1 2015/A2 2019; EN ISO 11137-2 2015; EN ISO 11137-3 2017. pH: 4–6. Treatment frequency: one application every 48 hours (h).

### 2.2. Preparation of Simulated Exudate Solutions

Pseudo-Wound Exudate solution (PWE) used for the analysis of the dressing properties was composed of 0.584% sodium chloride, 0.336% sodium hydrogen carbonate, 0.0298% potassium chloride, 0.0277% calcium chloride, 0.300% bovine albumin in purified water at pH 8.6. All chemicals used were purchased from Sigma-Aldrich (St. Louis, MO, USA). The Viscosity of the sample was 1.47 mPa/s. It was determined by using Rotational Viscometer (Visco Star plus) sample spindle L4 and speed of 6 rpm; at 37 °C.

Four Artificial Exudates (AE) were prepared for the lateral spread measurements. Test solution A was prepared according to EN 13726 standard. Briefly, 8.3 g of sodium chloride and 0.37 g of calcium chloride dihydrate were solubilized in 1L of deionized water. 400 µL of blue food dye (Permitted color E133) were added to the solution. Xanthan gum was solubilized in the colored Test solution A as thickening agent at different concentration (0.2–0.5–1.0–1.2% *w*/*v*) to obtain an increasing viscosity. The resulting suspension was stirred overnight until a clear solution was obtained. The viscosity of the AE was determined using a rotational viscometer (Rheomat 108, Contraves), at a temperature of 23 ± 2 °C, using a probe with a diameter of 30 mm.

### 2.3. Dressing Properties

#### 2.3.1. Wettability Test

The wettability was tested according to the ATCC 79 method with some modifications. In brief, the specimen (10 cm^2^) was placed 10 mm below the tip of the burette filled with PWE. A timer was started when one drop (50 µL) fell on the fabric surface and stopped when the drop lost its reflectivity. Each sample was tested for five drop sites and the average values reported.

#### 2.3.2. Dehydration Rate

The specimen (4 cm^2^) was immersed in PWE solution for 30 min. Then, the dressing was removed, weighed, and incubated in an oven at 37 °C for up to 48 h. At different time points (2–10–18–24–48 h), the dressing was weighed again (W1). The experiment was performed in triplicate, and the dehydration rate was calculated as reported in Equation (1) considering the weight of a non-imbibed dressing as background (W0). Data are represented in a mean ± standard deviation.
(1)$$Dehydration\;rate\left(\%\right)=\frac{\left(W0-W1\right)}{\left(W0\right)}100$$

#### 2.3.3. Water Vapor Transmission Rate (WVTR)

Circular specimens with a diameter bigger than the outer diameter of the test dish were first prepared by cutting. Then, 46 cm^3^ of PWE was put into a Petri dish with an inner diameter of 83 mm. The bottom layer of the specimen was fixed at the edge by using an adhesive tape to avoid the PWE evaporation. The distance between the surface of the solution and the underside of the specimen was 10 mm. Then, the Petri dish was put on a tilting plate with a rotation speed of 30 rpm on a thermostatic bath fixed at 37 °C up to 48 h. The dressing was weighed at the beginning and at different time points (2–24–48 h). The positive control was the bottle of the test solution that had no cover, and the negative control was the bottle of the test solution covered with parafilm. The experiment was performed in triplicate, and the WVTR was calculated as reported in Equation (2) and represented in a mean ± standard deviation.
(2)$$WVTR\left(\frac{g}{\left(m^{2}\;24\;h\right)}\right)=\frac{\left(M0-M1\right)}{\left(A\right)}$$

A = πd^2^/4 (d = 0.083 m) where M0 and M1 are the masses of the dish with water and fabric sample at the beginning and after time points (g); A is the evaporation area (m^2^); and d is the inner diameter of dish (m).

### 2.4. Simulated Exudate Solution Uptake

The specimen (4 cm^2^) was immersed in PWE and incubated at 37 °C up to 48 h. The dressing was weighed at the time point 0 and at 2, 24 and 48 h. PWE absorption of the samples (n = 3) was calculated based on Equation (3):(3)$$\%\;Uptake=\frac{\left(W1-W0\right)}{W0}100$$
where W1 is the weight of the hydrated dressing at different time point, and W0 is the initial dry weight of the dressing.

### 2.5. In Vitro PHMB Release

To assess the interaction of the ingredients delivered by the hydro-dispersible cream trough the triple layer system to the external surface, the release of PHMB from the MD was investigated by UV–Vis spectrophotometer. Briefly, the specimen (10 cm^2^) was immersed in purified water (1.3 mL/cm^2^), and put in a water bath at 37 °C. At appropriate time intervals, 1 mL aliquot was withdrawn and replaced with purified water. The absorbance of the aliquots was measured by UV–Vis spectrophotometer at the wavelength of 236 nm. The actual amount of PHMB released from the dressings was back calculated from the data obtained against a predetermined calibration curve of the hydro-dispersible cream containing 0.1% PHMB at serial dilutions. The experiment was performed in triplicate.

### 2.6. Antioxidant Capacity

The antioxidant activity was determined using the DPPH antioxidant assay [27] with slight modifications. Trolox was used as the reference compound. Briefly, 0.500 g of cream was mixed with 1.5 mL of Ethanol. The solution was vortexed for 10 min and mixed with 1.5 mL of 0.16 mM DPPH solution. The mixture was vortexed for 1 min, kept for 30 min in dark and then, the absorbance was measured at 517 nm in a PerkinElmer UV-Vis Spectrometer Lambda 20. The antioxidant capacity was calculated using the Equation (4):(4)$$\%\;Inhibition=\frac{\left(A0-A1\right)}{A0}100$$
where the A0 is the absorbance of the control (DPPH without sample), A1 is the absorbance of the test sample (the sample test and DPPH solution). Trolox was used as positive control.

### 2.7. In Vitro Cytotoxicity Assay

Human keratinocytes (HaCaT cells, American Type Culture Collection (Manassas, VA, USA) were cultured in Dulbecco’s Modified Eagle’s Medium (Lonza, Basel, Switzerland), supplemented with 10% fetal bovine serum, l-glutamine (2 mM), 100 unit/mL penicillin, 100 μg/mL streptomycin and maintained at 37 °C in a humidified atmosphere containing 5% CO_2_. The cell line has been authenticated by BMR Genomics (Padova, Italy). To evaluate the potential cytotoxicity of Rigenase, HaCaT cells were seeded in 96-well plates at a density of 5 × 10^3^ cells/well in 200 µL of medium. After 24 h incubation, 6 and 60 µg/mL of Rigenase were added to the wells at and incubated for 48 h at 37 °C in 5% CO_2_. Cell viability was assessed by the colorimetric method based on [3-(4,5-dimethylthiazol-2-yl)-5-(3-carboxymethoxyphenyl)-2-(4-sulfophenyl)-2H-tetrazolium, inner salt; MTS] (Promega Corporation, Madison, WI, USA). The optical density of the chromogenic product was measured at 490 nm. Data were expressed as percentage of viable cells compared to untreated cells.

### 2.8. MD Stability by Fourier Transform Infrared Spectroscopy

The dressing properties were characterized with Fourier transform infrared spectroscopy (FTIR). The specimen (4 cm^2^) was immersed in PWE (1.3 mL/cm^2^) for 48 h and dry in an oven overnight at 37 °C. FTIR spectra ranging between 4000 and 450 cm^−1^ were recorded Using a PerkinElmer Fourier FTIT/FIR spectrometer, Spectrum 3, equipped with a Universal ATR accessory containing a Thallium Bromo-iodide Crystal. All the data were analyzed with the PerkinElmer IR Enhanced security software, Spectrum 100.

### 2.9. Antimicrobial Study

#### 2.9.1. Bacterial Strains

Each microorganism (Gram+ *S. aureus* ATCC 6538—Microbiologics, St. Cloud, MN, USA and Gram- *K. pneumoniae* ATCC 13883—Thermo Fisher Scientific Waltham, MA USA) was streaked onto TSA 90 mm plates (Tryptic Soy Agar—Merck—prepared according to the manufacturer’s instructions) for 18–24 h at 30–35 °C. Subsequently, one isolated colony was picked from the plate and suspended in NaCl 0.9% and adjusted to equal turbidity of 0.5 McFarland standard. For each microorganism, 1 mL of suspension was added to 9 mL of PWE suspension (1.5–3 × 10^6^ CFU/mL).

#### 2.9.2. Antimicrobial Properties of the Dressings: Test 1

Films of 12.5 cm^2^ of FPA were prepared. A film of triple layer fabric without cream was used as control. For each microorganism, each film surface was then inoculated with 50 μL of PWE suspension and covered with a plastic film to prevent drying, put in empty Petri dishes containing a piece of paper saturated with PWE to maintain high humidity and incubated in thermostatic bath fixed at 37 °C with agitation for 48 h. At intervals of 2, 24 and 48 h, bacterial burden was determined by transferring films to a 50 mL conical tube filled with 10 mL Sterile neutralizer (3.6 g/L Potassium Dihydrogen Phosphate; 7.2 g/L Disodium Hydrogen Phosphate Dihydrate; 4.3 g/L Sodium Chloride; 1 g/L Peptone; 30 g/L Tween 80; 30 g/L Saponin; 0.5 g/L Histidine; 3 g/L Lecithin; 5 g/L Sodium Thiosulfate Pentahydrate) and agitating 5 min to stop the PHMB effect. 1 mL of the resulting solution was serially diluted, the number of surviving bacteria was counted on agar plate (Tryptic Soy Agar—Merck—prepared according to the manufacturer’s instructions) incubated for 24 h at 37 °C and the log reduction was calculated by subtracting time point log_10_ count on control.

#### 2.9.3. Antimicrobial Properties of the Dressings: Test 2

The antibacterial activity was evaluated by the determination of the antimicrobial activity of immobilized antimicrobial agents under dynamic contact conditions (ASTM E2149-01) with some modifications. Briefly, for each microorganism, the specimen (1 cm^2^) was placed in 5 mL conical tube filled with 3 mL of a diluted PWE suspension (approx. 3.0 × 10^5^ CFU/mL). The triple layer fabric without cream (1 cm^2^) was used as control. Tubes were vigorously shaken and incubated in thermostatic bath fixed at 37 °C with agitation for 48 h. At intervals of 2, 24 and 48 h, bacterial burden was determined by transferring 1 mL of PWE suspension to a 15 mL conical tube filled with 9 mL Sterile neutralizer (as described above) and agitating for 5 min to stop the PHMB effect. 1 mL of the resulting solution was serially diluted, the number of surviving bacteria was counted on TSA agar plate, incubated for 24 h at 37 °C and the log reduction was calculated by subtracting time point log_10_ count on control.

The experiments were run in duplicate. The appropriate controls such as media sterilization control, viability growth control, and neutralization control were performed.

### 2.10. Lateral Spread Measurement

The lateral spread was tested by our novel in vitro method for advanced wound dressing. Briefly, the specimen was fixed between two grilled supports. At the upper level, 0.5 mL of AE was introduced in a cylinder (diameter 2.5 cm). One minute after applying the AE, the upper grille was removed. The diffusion area was determined by images processed with the analysis free ImageJ software.

Then, we conducted an in vitro test to reproduce the clinical condition of a highly viscous purulent exudate wound covered by a dressing under elastic compression bandage.

A simulated surface lesion of 14.5 cm^2^ with 1.5 cm of maximum depth, 11 mL capacity, filled with 11.5 g of AE at 1.2% (*w/v*) was covered with the dressing. Then a cotton gauze, a plexiglass tablet (5 × 5 cm^2^) and a weight (40 mmHg pressure) were place on the simulated surface lesion to simulate the elastic compression.

After 72 h, the weight was removed and images of both the dressing, and internal and external lateral diffusion were taken and measured. The test was repeated in triplicate.

## 3. Results

### 3.1. Triple Layer System

As an advanced absorbent dressing, FPA consists of a PET triple layer bonded fabric system soaked with a hydro-dispersible cream. Notably, this formulation is enriched with an aqueous extract of *Triticum vulgare* that acts as a soothing and calming agent. The whole system allows gaseous exchange between wounded tissue and external environment, absorbs a high volume of wound exudate, and prevents bacterial infection. Figure 1 is a schematic representation of the triple layer system.

In detail, the three layers differ in technical properties (Table 1) and consist of a wound contact bottom layer (Monofilament Fabric), an absorbent layer (Monofilament Knitted Fabric), and a protective top layer (Monofilament Fabric) assembled by ultrasonic welding bonding without added adhesive.

Thus, even though PET is recognized as a moderately hydrophobic polymer [28], the difference in structure and morphology of the inner reservoir fabric and the outer layer imbibed with cream allows the optimal breathability necessary for such a complex environment as the chronic wound.

### 3.2. Dressing Properties

Wettability is a physical property indicative of the affinity of materials with a liquid. A shorter wetting time indicates a better wettability. This mainly depends on the support material used, the arrangement of layers and the properties of the applied formulation. The specimen’s wettability, tested with PWE, was 12 ± 0.8 s. Ideal support should show relatively short wettability times (5–60 s) to allow a faster exudate absorption and to avoid its interaction with the adjacent unscathed area [29,30]. So, we can speculate that the external openwork fabric of our compound contacts directly with the exudate and quickly guides the liquid over.

To further corroborate our hypothesis, we investigated the dehydration properties of the dressing (Figure 2a). Interestingly, we found out that dehydration rate was very high in the first 10 h (82.565%). After 18 h, it achieved the 90%. This turned out in reducing both the risk of skin maceration due to an excessive accumulation of liquids, and on the other side an adhesive and painful effect due to a too dry dressing [31,32].

Moreover, an ideal system should allow an appropriate moist environment during the wound healing progress. This could be provided by good management of the WVTR. Indeed, a too high WVTR can lead to rapid dehydration of the injured skin, conversely, a relatively low WVTR can increase the risk of maceration and infection of the wound and tissues [29]. As reported in Figure 2b, FPA showed a WVTR of 1944.828 g/m^2^/24 h. WVTR values between 100–3300 g/m^2^/24 h are suitable to maintain optimal moisture content, to promote cell proliferation and functions without causing dehydration [33]. However, different wound types require different WVTR dressings. The average WVTR for normal skin is 204 g/m^2^/24 h while for wounds varies between 279 (first-degree burn) and 5138 g/m^2^/24 h (granulation tissue) [34,35]. The optimal maximum value of the WVTR for the healing of deep wounds such as ulcers and sores in a humid environment is defined between 840 and 2500 g/m^2^/24 h [33]. So, our system shows a good balance between wettability and breathability for the treatment of hard-to-heal chronic wound.

### 3.3. PWE Uptake

After being wetted, the yards began to absorb great amounts of exudate. This mechanical action is essential for the wound healing process especially in chronic high-exuding stages. Indeed, it promotes the wound debridement, the diffusion of signaling molecule nutrients, and consequentially the restoration of a microbiologically regulated environment [36]. To better reproduce the pathological condition of a high-exuding injured skin the absorption property of our dressing was investigated in a PWE (1.3 mL/cm^2^) at pH 8.6 37 °C up to 48 h.

Interestingly, Figure 3a shows an absorption already in the first 2 h. A plateau of 50% *w*/*w* was reached out between 24–48 h. Simultaneously, the cream entrapped in the fabric network begins to disperse into the surrounding environment.

### 3.4. In Vitro PHMB Release

To better understand this mechanism, we monitored the release of the hydro-soluble compound PHMB used as tracer in purified water (1.3 mL/cm^2^) at 37 °C up to 48 h (Figure 3b). The PHMB release was investigated by UV-Vis at 236 nm as reported by Mann et al. [37]. It reached out a 60% in the first 2 h. However, it was prolonged and steady between 24–48 h. This behavior indicates that the ingredient availability will be on the wound site for 2 days, which is the prime requirement for dressing application [38].

### 3.5. Antioxidant Capacity and In Vitro Cytotoxicity Assay

The presence of Rigenase (*Triticum vulgare* extract) allows FPA to exhibit an antioxidant activity attested at 72% by DPPH assay and supported by soothing and emollient properties of PEGs as consistency factors.

Moreover, to assess its in vitro biocompatibility, we investigated Rigenase citotoxicity at two different concentrations (6 and 60 µg/mL) on HaCaT keratinocytes. Notably, after 48 h the cell viability was not affected (Figure 4) demonstrating that this natural compound is no toxic in an in vitro 2D model.

### 3.6. Fourier Transform Infrared Spectroscopy (FTIR)

The FT-IR spectrum of the triple layer fabric in Figure 5 shows absorptions usually known as the fingerprint of PET material.

Particularly, the strong band at 1711.31 cm^−1^ is associated with the stretching vibrations of ester groups, the absorptions at 1239.70 cm^−1^ and 1094 cm^−1^ are due to stretching of C-O bond, and 722 cm^−1^ for the bending of C-H bond [39,40]. A FT-IR analysis was recorded also on FPA imbibed in PWE at pH 8.6 for 48 h. Its spectrum contains the same complex pattern of absorption of the triple layer wet fabric. Furthermore, this second spectrum looks very clear, and it doesn’t show any other peaks due to new functional groups. To deeply investigate how much the result of the two analyses are superimposable, we used the “Compare” function of the software which allows us to relate the two spectra, giving a 99.06% of overlap. Thus, we can speculate that FPA doesn’t degrade nor release subproducts once in contact with such a complex environment as a chronic wound.

### 3.7. Antimicrobial Activity

The antimicrobial activity of the dressing was determined against the Gram-positive *S. aureus* and the Gram-negative *K. pneumoniae* in two different tests and the results are presented in Table 2 and Table 3. As can be seen in Table 2 and Table 3, under the test conditions, the fabric material itself did not cause a significant reduction in bacterial viability. All the experiments exhibited antimicrobial activity for both *S. aureus and K. pneumoniae*. Reductions in viability were observed after less than 2 h. Culturable *K. pneumoniae* levels at time zero declined below the detection limit of the enumeration technique (<1 Log_10_ CFU/mL) and no increase in *K. pneumoniae* CFU/mL was detected in subsequent time points for both tests. At time zero, in presence of FPA, a bacteriostatic effect was observed for *S. aureus*, but already after 24 h bacterial survival of *S. aureus* was affected throughout the period of application.

The antimicrobial activity was tested by simulating a wound in vitro. The test was performed by polluting PWE with the Gram-negative *K. pneumoniae* and the Gram-positive *S. aureus*, then applying the dressing in dynamic conditions of humidity and temperature up to 48 h. After 2, 24, and 48 h the number of viable cells was determined.

We observed the ability of FPA to contain the microbial load in 48 h of simulated dressing application. In particular, for the microorganism *K. pneumoniae,* the effect is already noticeable at time zero, considering that the simulated exudate seems to be a particularly favorable environment for the exponential growth of this microorganism.

The PWE keeps the growth of the *S. aureus* microorganism constant over 48 h, but, after applying the dressing, already at time zero the growth of the microorganism begins to decrease, demonstrating the ability of FPA to manage the microbial load in conditions mimicking wound in vitro.

### 3.8. Vertical Diffusion

The AE at different concentration of xanthan gum showed the typical rheological profile of a non-Newtonian fluid, where the viscosity depends on shear rate changes (D). Table 4 shows the viscosity values of AE with increasing amount of xanthan gum. It varied with varying of D. Since it was not possible to determine an absolute viscosity value to compare the different AEs, it was chosen a D value of 1290 s^−1^ measurable for all samples.

The ability to drive the absorbed exudate towards the external surface (vertical diffusion) is essential for an advanced dressing to avoid damaging of the perilesional skin.

Ideally, a proper management of the exudate is represented by an external to internal diffusion area ratio ≥1. Since there isn’t a univocal study to evaluate the reliability of this parameter, we developed a specific device to set up a novel test (2.10 Materials and Methods Appendix A) able to reproduce in vitro the clinical management of the wound dressing.

Figure 6 shows a visible reduction of lateral diffusion on the inner surface of the dressing at increasing viscosity AE values. No vertical diffusion was observed for 1% and 1.2% AE.

Furthermore, to reproduce in vitro the clinical condition of a generic chronic wound medication we performed the lateral spread test simulating a bandage elastic compression on an ulcer with high viscosity exudate (1.2% p/v) (Appendix A).

Table 5 shows the external (D1) and the internal (D2) lateral diffusion values of the dressing, the percentage diffusion area (considering 100 cm^2^ as the total surface dressing area) and the D1/D2 ratio.

The internal lateral diffusion of exudate occupied just the 18% of the total area. The D1/D2 ratio is> 1. This data demonstrated that FPA could manage the exudate production through the phenomenon of the vertical absorption.

Furthermore, we picked up the liquid remained in the wound bed. The difference between the initial volume and the one retained by the dressing in the bed of the simulated wound, was assessed to be 43%. In the literature, it is reported that the amount of exudate produced by a leg ulcer varies from 0.43 to 0.63 g/cm^2^/24 h^3^ [41] Therefore, considering the size of the ulcer simulated (about 14.5 cm^2^), FPA may be able to manage the exudate produced in 48 h. Figure 7 shows absorption the aqueous component of the exudate while the hydrophobic material is concentrating in the inner surface of the dressing.

## 4. Discussion

During the last decades the treatment of hard-to-heal wound reached a great importance to improve human’s quality of life and extend its expectancy. Even though the big progress in understanding the mechanisms related to the healing progress, the heterogeneity, and the irregularities of such a complex physiological process represent major challenges. Thus, there is still an urgent need for an adequate and well-planned management of chronic wounds.

For many years, traditional dressings as cotton wool, tulle, or lint bandages and single-layer gauzes have been extensively used just for covering the wound itself [42]. These dressings ensure a clean wound and prevent further injuries from the external environment. However, they are not designed to promote the optimal condition for a suitable recovery in chronic conditions. Indeed, traditional dressings absorb a high amount of blood and exudate especially in hard-to-heal wounds leading to dried surface and formation of clots [16]. Furthermore, the fibers easily stick to the damaged tissue causing a painful removal of the dressing [43]. Winter et al. was the first to highlight the importance of the moisture retention capability. Notably, they demonstrated that moist environment speeds the wound healing up to 50% compared to dry, open to air conditions [44]. This is due to the autolytic debridement necessary for the healing progress, that is favored by a limited amount of exudate retained on the wound [16].

To overcome these limitations and to recreate the ideal wound healing environment, over the years advanced dressings have been designed and developed.

The modern medications, currently used in clinical practice, are made of semi-permeable and highly absorbent materials assembled in a way that creates a moist, warm environment impermeable to microorganisms, removes excess exudates, provides protection to periwound area and can conform to wound shape [31,45]. In the last decades several studies reported the invention and development of three-dimensional, multi-layer wound-dressing systems [46]. An example, Gupta et al. [38] developed new dressings with non-woven polyester fabric via freeze-drying approach. They loaded Tetracycline hydrochloride drug along with curcumin and aloe vera on these dressings to obtain a controlled antimicrobial activity. Interestingly, the cumulative release of drug from the dressings increased with increasing immersion time and, even though it was very fast in the first 10 h, it continued up to 50 h. Moreover, WVTR was found to be in the range 2000–2500 (gm^−2^ day^−1^) and PBS uptake higher than absorbency 900%. Noteworthy, few studies reported the development of compression bandages and absorbent dressings based on spacer fabrics. Yang et al. [47] proposed new types of dressings produced by using a single polyester/spandex (100D/40D) yarn to knit the two outer layers, and a 32S/2 bleached cotton or Tencel yarn to knit the spacer layer with a connecting distance of 4 needles, respectively. They further modified these dressings by covering a polyurethane or a polystyrene electro spun nanofibrous membrane onto their outer layer surface. The results compared to Alginate and Foam dressings showed that these spacer fabric-based devices could absorb a large amount of fluid in a short period (wettability was no higher than 30 s) and demonstrated a good breathability with a WVTR around 900 (g/(24 h m^2^)).

Tong Shuk-fan et al. [48] reported the application of warp-knitted spacer fabrics for advanced wound dressing. Notably, the three-dimensional nature of this structure provided a proper moist environment with low evaporation and rapid absorption of a large amount of fluids. Similarly, Yang et al. [29] developed an absorbent wound dressing based on spacer fabric. In details, this three-dimensional system was assembled with two external hydrophobic surfaces and an absorbent middle layer. The spacer fabric had a remarkable Moisture Vapor Transmission Rate (MVTR) of 2069 (g/(24 h m^2^)), which was beneficial for keeping a moist environment and to accelerate wound healing. Furthermore, the wettability (23.81 s) and the high absorbency (30.13 g/100 cm^2^) suggested a rapid and efficient absorption of the exudates from the wound.

Despite the recent advance in chronic wound management, there is still a need to understand the detailed pathophysiology of wounds and the healing process. Furthermore, specific in vitro and in vivo characterizations are needed to fill the gap between technical dressing properties and clinical practice.

To address these needs, we developed Fitostimoline Plus Garze ADVANCE, a new topical medical device. Given the combination of the advanced PET non-woven triple layer imbibed with a special formulation based on Rigenase, it forms a protective barrier against the external environment, creating favorable conditions for a rapid and correct re-epithelizing action on the skin and helps to keep the micro-environment under control for the management of hard-to-heal chronic wounds.

In detail, Rigenase is a specific aqueous extract obtained by Farmaceutici Damor from the natural source of *T. vulgare*. It exhibits a moisturizing action and a scavenging effect toward free radicals, thus pointing to its relevant antioxidant activity [25]. At date, several medical devices—under brand name Fitostimoline—boast Rigenase as the fundamental ingredient that gives the additional ability to create a favorable condition for a rapid and correct re-epithelization in the wound healing field.

The results from our technical evaluation of dressing properties showed that the triple layer system is effective. It does not cause occlusion and above all it is capable to manage high amount of exudate driving it vertically from the wound site to the upper layer.

The synergically activity of the PET fabrics and the hydro-dispersible cream imbibed in the reservoir system allows a good balance between hydrophilicity and hydrophobicity thanks to the special formulation of the cream with Rigenase. The dressing acts as a mechanical drainage, detracting potentially infected exudate during the application period of 48 h. At the same time the PET in presence of cream and alkaline conditions remains undamaged and thanks to the ultrasonic welding it leaves no residue, and it is easy to remove. Moreover, to simulate the clinical treatment of a chronic skin lesion we performed an in vitro analysis of the triple layer system lateral exudate diffusion by our novel in house test. Interestingly, the lateral diffusion value was D1/D2 > 1 demonstrating excellent vertical transmission of the exudate. Standing this point, we can speculate that our novel wound dressing is an ideal candidate for the treatment of chronic lesion in clinical practice. As a confirm, a recent work investigated the clinical efficacy and tolerability of Fitostimoline Plus Garze ADVANCE for the treatment of chronic skin lesion. Interestingly, the authors concluded that the dressing demonstrated a high ability to stimulate the resumption of repair phenomena in lesions with little spontaneous tendency to repair [49].

## 5. Conclusions

A medical device for topical use is needed to enhance the drug therapy administered and to accelerate healing. Our study demonstrates the effectiveness of Fitostimoline Plus Garze ADVANCE in fulfilling this role.

Our results indicate that Fitostimoline Plus Garze ADVANCE has a non-occlusive effect for the entire duration of the application (48 h), guaranteeing favorable humidity conditions. It counteracts the proliferation of both Gram+ and Gram- bacterial agents, proving to be a useful tool for containing the microbial load. The special combination of the triple imbibed layer of Rigenase-based cream makes the product able to remove the infected exudate from the environment, at the same time it exerts a barrier effect by preventing the external contamination and containing Rigenase, it could also favor rapid re-epithelialization. The 48 h treatment schedule makes it ideal for the management of hard-to-heal chronic wounds.

## Figures and Tables

**Figure 1 bioengineering-10-01022-f001:**
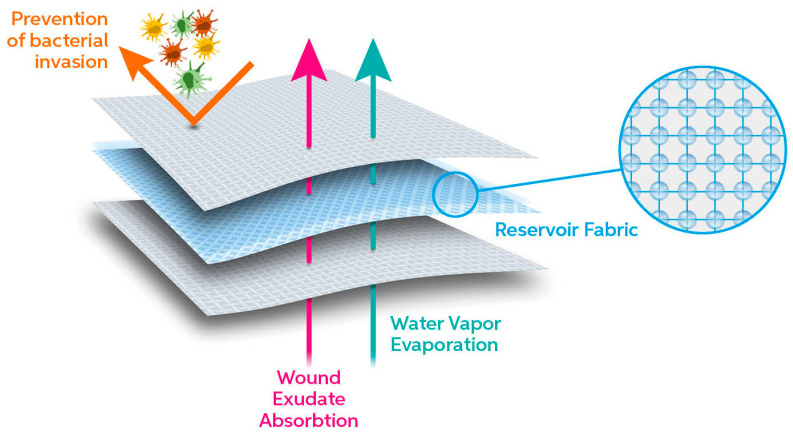
Schematic illustration of the triple layer fabric system and its function.

**Figure 2 bioengineering-10-01022-f002:**
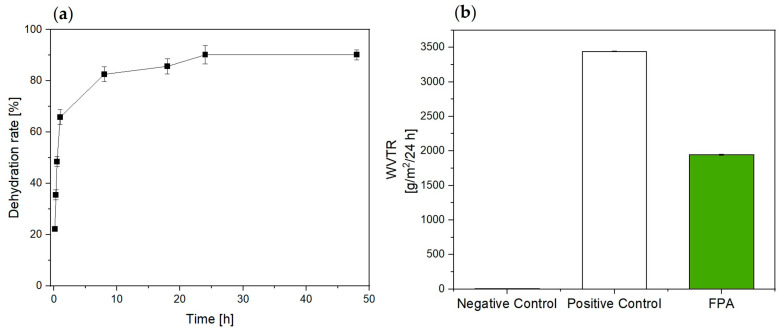
(**a**) Dehydration rate and (**b**) WVTR of FPA compared to an occlusive (Negative Control) and non-occlusive system (Positive Control).

**Figure 3 bioengineering-10-01022-f003:**
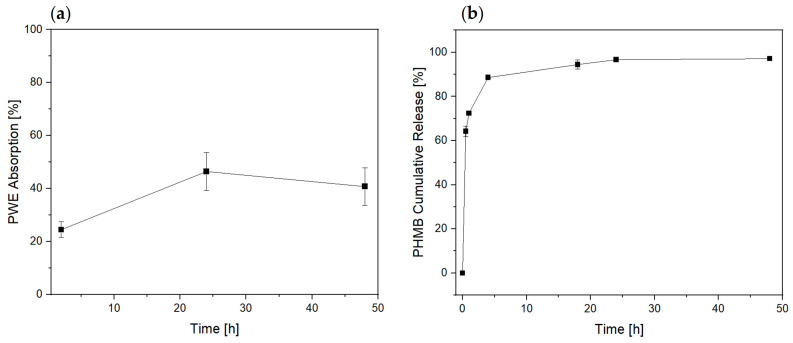
(**a**) PWE absorption of dressing; (**b**) PHMB Cumulative Release profile tested at 37 °C up to 48 h.

**Figure 4 bioengineering-10-01022-f004:**
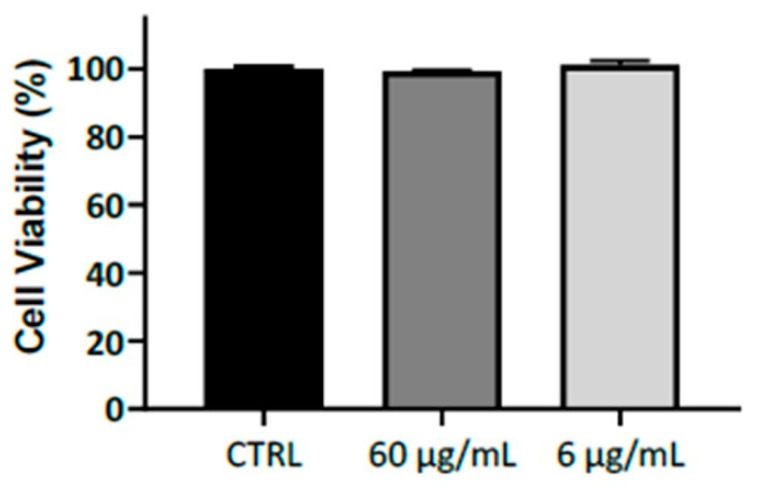
Effect of Rigenase at 6 and 60 µg/mL on HaCaT keratinocytes viability. Results are expressed as percentage of cell viability relative to untreated control cells (CTRL). Data are shown as means ± standard error (n = 6).

**Figure 5 bioengineering-10-01022-f005:**
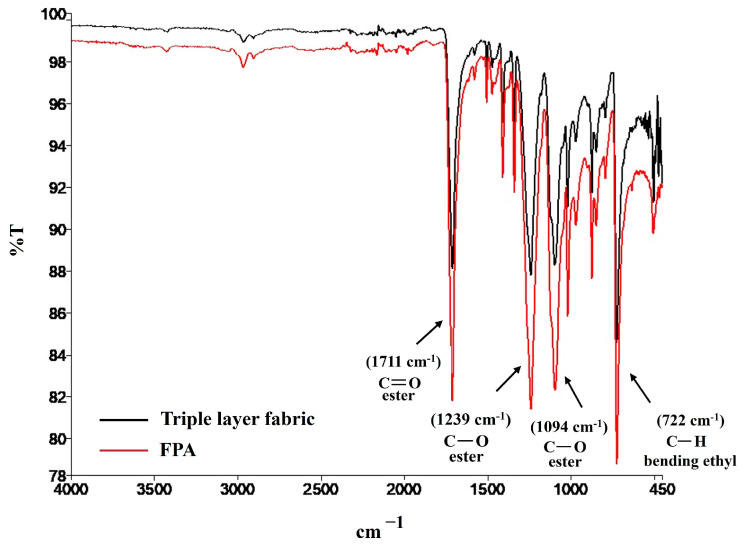
The FT-IR spectrum of the triple layer wet fabric at time point 0 compared to FPA tested at 37 °C up to 48 h in alkaline conditions.

**Figure 6 bioengineering-10-01022-f006:**
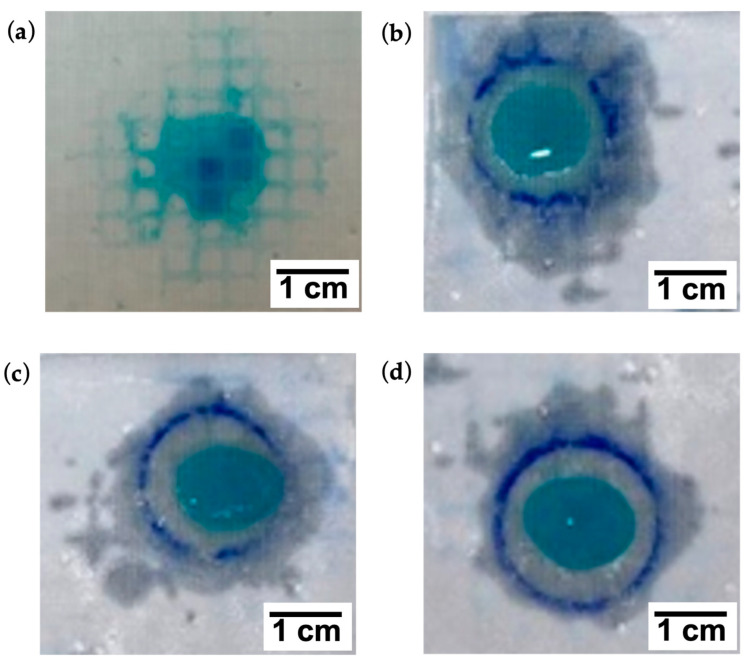
Lateral spread of FPA treated with 0.2% (**a**); 0.5% (**b**); 1% (**c**) and 1.2% (**d**) AE.

**Figure 7 bioengineering-10-01022-f007:**
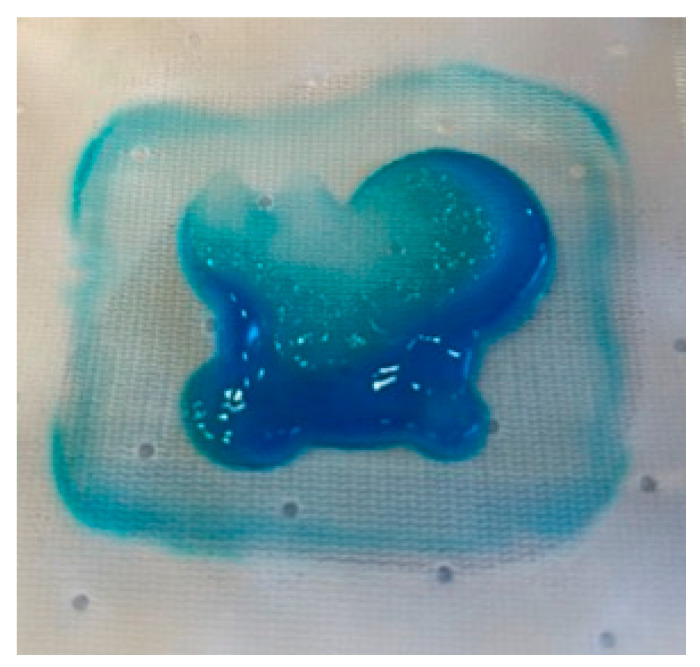
AE (1.2% p/v) exudate concentrated on the inner surface of the dressing.

**Table 1 bioengineering-10-01022-t001:** Technical Parameters of Monofilament Fabric and Monofilament Knitted Fabric (data provided by Sefar Group).

Technical Parameters	
	**Monofilament Fabric**
Mesh Opening [µm]	190 ± 15
Mesh Count [n/cm]	44
Open Area [%]	70
	**Monofilament Knitted Fabric**
Gauge Knitt Course [n/cm]	11
Gauge Knitt Wale [n/cm]	11

**Table 2 bioengineering-10-01022-t002:** Log_10_ bacteria burden (CFU/mL) and Log_10_ reduction under the test 1 and test 2 conditions for *K.pneumoniae* during application time of FPA.

*K. pneumoniae*			
	Control	FPA	Log_10_ Reduction
Test 1			
T0	4.42	<1.00	>3.42
2 h	4.70	<1.00	>3.70
24 h	5.00	<1.00	>4.00
48 h	5.79	<1.00	>4.79
Test 2			
T0	3.18	<1.00	>2.18
2 h	4.00	<1.00	>3.00
24 h	4.20	<1.00	>3.20
48 h	5.60	<1.00	>4.60

**Table 3 bioengineering-10-01022-t003:** Log_10_ bacteria burden (CFU/mL) and Log_10_ reduction under the test 1 and test 2 conditions for *S. aureus* during application time of FPA.

*S. aureus*			
	Control	FPA	Log10 Reduction
Test 1			
T0	4.75	2.80	1.95
2 h	4.34	2.75	1.59
24 h	4.60	<1.00	>3.60
48 h	4.10	<1.00	>3.10
Test 2			
T0	4.30	2.50	1.80
2 h	4.34	<1.00	>3.34
24 h	4.80	<1.00	>3.80
48 h	3.80	<1.00	>2.80

**Table 4 bioengineering-10-01022-t004:** AE viscosity and lateral spread at the internal surface of the dressing.

AE Concentration % (*w/v*)	Viscosity (mPa·s)	Lateral Spread (cm^2^)
0.2	9.0	13.69
0.5	1.6	9.01
1.0	2.9	5.84
1.2	3.7	3.69

**Table 5 bioengineering-10-01022-t005:** External and internal lateral spread % and D1/D2 ratio. Measurements in triplicates are express as Mean Average.

D1 (cm^2^)	D2 (cm^2^)	Lateral Spread % (Outer Side)	Lateral Spread % (Inner Side)	D1/D2
		Mean ± DS		
17.7 ± 0.9	15.5 ± 0.6	20.6 ± 1.1	18.1 ± 0.7	1.1 ± 0.1

## Data Availability

The data that support the findings of this study are available from the corresponding author upon reasonable request.

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
