# Peer review of "Technical Evaluation of a New Medical Device Based on Rigenase in the Treatment of Chronic Skin Lesions"

_bioengineering, 2023, doi:10.3390/bioengineering10091022_

Round 1
Reviewer 1 Report
This manuscript is well written and can be accepted for publication after the authors provide sufficient responses to the following comments:
1. FTIR spectra in Figure 5 should be indexed.
2. The quality of Figure 7 should be improved
3. The discussion section is too short, the authors need to provide comprehensive discussion.
Author Response
Comments and Suggestions for Authors:
This manuscript is well written and can be accepted for publication after the authors provide sufficient responses to the following comments.
We thank the Reviewer 1 for all comments. We hope that the revised version of the manuscript will address his requirements.
Point 1: FTIR spectra in Figure 5 should be indexed.
Response 1: Following the suggestion of Reviewer 1 Figure 5 was indexed.
Point 2: The quality of Figure 7 should be improved
Response 2: We add an additional file with the Supplementary data relative to the lateral diffusion tests performed and included the representative image of the diffusion test as Figure S1.
In the main body of the manuscript,we decided to describe the setting up of the novel test in the 2.10 Materials and Methods Section. Please see reference “Mennini N., Fiani S., Pacini F., Bellingeri A., De Vita F, Petrella F., Greco A. Development of an in vitro method for measuring the lateral spread in advanced wound dressings. 33th Conference of European Wound Management (Association (EWMA), Milano, 3-5 maggio 2023, e-Poster Prize Winner 2023”)
Point 3: The discussion section is too short, the authors need to provide comprehensive discussion
Response 3: Following the suggestion of Reviewer 1 we fully revised the Discussion section

Reviewer 2 Report
Authors have conducted this in vivo study to investigate the technical and physical properties of a novel advanced PET non-woven triple-layer gauze imbibed with a cream containing Rigenase, a Triticum vulgare aqueous extract used for the treatment of skin injuries.
Overall, I do believe that their work is a significant contribution to this field, and their extensive investigations on this commercial patch for the treatment of chronic skin lesions, do bring value to better understanding/predicting the clinical outcomes of this product.
I suggest accepting this manuscript after minor revisions. The discussion section of this paper needs serious reconstruction and improvements. I have listed my concerns and suggestions as followed:
Abstract:
The abbreviation “PET” and any other abbreviations in the abstract and the main manuscript need to have their full form detailed in the text, when being used for the first time.
The abstract must be structured as Introduction/Background/Objectives, Methods and Materials, Results, and Conclusions. I do not understand why the words “Herein” and “The” are in bold?
Keywords:
The word “medical device” is not a specifically-related word for this paper and it is so general and therefore unnecessary. I suggest adding the word “Rigenase” instead.
Introduction:
In the last line of the first paragraph “[cuts, surgical incisions, metabolic dis- orders, ulcers] [4].”, if the authors want to name a few examples they can use this format: (e.g., … and …). And if they want to refer to a specific subject/item they can use this format: (i.e., …).
The references are relatively newly published and all related to the subject of the paper.
Methods and Materials:
All of the methods and different steps of the study are described and detailed adequately. The thought process behind the technical decisions is not vague.
Results:
The quality of the pictures and illustrations used in Figure 1 and figure 6 can be significantly improved. Specifically for Figure 6 where the numbers of the ruler are not so easily readable.
Authors need to clearly indicate and detail the descriptions for the negative control group and the positive control group of their study in the figure legend of Figure 2.
In Figure 3, why have authors chosen to only report the results of the first 50 hours of their examination and not longer? Especially for the cumulative release examination. And also, was there a specific reason that authors chose to report these results in percentile?
In Figure 5, I believe that it would be beneficial for the readers and reviewers if authors indicated the specific bond responsible for each of the peaks and/or lowest points of their FT-IR examination. You can better understand my comment by taking a look at the FT-IR result figures of this article: 3D‐printed MgO nanoparticle loaded polycaprolactone β‐tricalcium phosphate composite scaffold for bone tissue engineering applications: In‐vitro and in‐vivo evaluation
Discussion:
I believe that the discussion of this paper is relatively short and limited. Given the novelty of their product and their respected application methods, I believe there are way more in vitro, in vivo, and clinical studies that can be discussed here. Additionally, I believe since the final goal of this study is to design a product for clinical use, authors must definitely discuss the need for this type of product in clinic, way more.
The main issue with the discussion section of this paper is the fact that it only has one reference and the rest of the discussion is just the authors’ report of their results and detailing their personal opinion on the matter. The discussion of this in vitro study not only has to include the most similar studies to compare their results with, but it also has to refer to a couple of in vivo and human clinical studies to draw out the indication and necessity, and novelty of this in vivo study.
The discussion section of this paper needs to be significantly improved with much-needed newly-published studies.
Author Response
Comments and Suggestions for Authors:
Authors have conducted this in vivo study to investigate the technical and physical properties of a novel advanced PET non-woven triple-layer gauze imbibed with a cream containing Rigenase, a Triticum vulgare aqueous extract used for the treatment of skin injuries. Overall, I do believe that their work is a significant contribution to this field, and their extensive investigations on this commercial patch for the treatment of chronic skin lesions, do bring value to better understanding/predicting the clinical outcomes of this product.
I suggest accepting this manuscript after minor revisions. The discussion section of this paper needs serious reconstruction and improvements.
We thank the Reviewer 2 for all comments. All the general and technical suggestions. We hope that the revised version of the manuscript will address his requirements.
Abstract: The abbreviation “PET” and any other abbreviations in the abstract and the main manuscript need to have their full form detailed in the text, when being used for the first time. The abstract must be structured as Introduction/Background/Objectives, Methods and Materials, Results, and Conclusions. I do not understand why the words “Herein” and “The” are in bold?
Response 1: Following the suggestion of Reviewer 2 we revised all the terms and abbreviations. We modified the Abstract adding more detailed about the Background, the Aim of Work, the Methods and the Results. However, we applied the template required by the journal.
Keywords: The word “medical device” is not a specifically-related word for this paper and it is so general and therefore unnecessary. I suggest adding the word “Rigenase” instead.
Response 2: Following the suggestion of Reviewer 2 we revised the keywords.
Introduction: In the last line of the first paragraph “[cuts, surgical incisions, metabolic dis- orders, ulcers] [4].”, if the authors want to name a few examples they can use this format: (e.g., … and …). And if they want to refer to a specific subject/item they can use this format: (i.e., …).
Response 3: Following the suggestion of Reviewer 2 we fixed all the sentences.
Results: The quality of the pictures and illustrations used in Figure 1 and figure 6 can be significantly improved. Specifically for Figure 6 where the numbers of the ruler are not so easily readable.
Authors need to clearly indicate and detail the descriptions for the negative control group and the positive control group of their study in the figure legend of Figure 2.
In Figure 3, why have authors chosen to only report the results of the first 50 hours of their examination and not longer? Especially for the cumulative release examination. And also, was there a specific reason that authors chose to report these results in percentile?
In Figure 5, I believe that it would be beneficial for the readers and reviewers if authors indicated the specific bond responsible for each of the peaks and/or lowest points of their FT-IR examination. You can better understand my comment by taking a look at the FT-IR result figures of this article: 3D‐printed MgO nanoparticle loaded polycaprolactone β‐tricalcium phosphate composite scaffold for bone tissue engineering applications: In‐vitro and in‐vivo evaluation.
Response 4: Following the suggestion of Reviewer 2 we improved the resolution of Figure 1. We modified Figure 5 by adding a readable scale bar with Image J Software.
Caption of Figure 2 was modified according to Reviewer 2 suggestion.
Experiments reported in Figure 3 were performed up to 48 hours. The rationale was to simultaneously monitoring the Exudate Absorption and the Polyhexamethylene biguanide release, in accordance with the time frame of application reported in the leaflet of the Medical Device. Also, the set-up of the experiments and the elaboration were done in accordance with several papers we used as our following references:
- Gupta, Bhuvanesh, Roopali Agarwal, and M. Sarwar Alam. "Antimicrobial and release study of drug loaded PVA/PEO/CMC wound dressings." Journal of Materials Science: Materials in Medicine 25 (2014): 1613-1622.
- Lumbreras-Aguayo, Angélica, et al. "Poly (methacrylic acid)-modified medical cotton gauzes with antimicrobial and drug delivery properties for their use as wound dressings." Carbohydrate polymers 205 (2019): 203-210.
- Khabbaz, Bahareh, Atefeh Solouk, and Hamid Mirzadeh. "Polyvinyl alcohol/soy protein isolate nanofibrous patch for wound-healing applications." Progress in biomaterials 8 (2019): 185-196.
Following the suggestion of Reviewer 2 we revised Figure 5.
Discussion: I believe that the discussion of this paper is relatively short and limited. Given the novelty of their product and their respected application methods, I believe there are way more in vitro, in vivo, and clinical studies that can be discussed here. Additionally, I believe since the final goal of this study is to design a product for clinical use, authors must definitely discuss the need for this type of product in clinic, way more. The main issue with the discussion section of this paper is the fact that it only has one reference and the rest of the discussion is just the authors’ report of their results and detailing their personal opinion on the matter. The discussion of this in vitro study not only has to include the most similar studies to compare their results with, but it also has to refer to a couple of in vivo and human clinical studies to draw out the indication and necessity, and novelty of this in vivo study. The discussion section of this paper needs to be significantly improved with much-needed newly-published studies.
Response 5: We thank the Reviewer 2 for the comment about the discussion section. The paragraph was fully revised. We added literature references and example from the most representative work describing the technical properties of dressing applied in the wound healing field. Please see:
- Gupta, Bhuvanesh, Roopali Agarwal, and M. Sarwar Alam. "Antimicrobial and release study of drug loaded PVA/PEO/CMC wound dressings." Journal of Materials Science: Materials in Medicine 25 (2014): 1613-1622.
- Yang Y, Bechtold T, Redl B, Caven B, Hu H. A novel silver-containing absorbent wound dressing based on spacer fabric. Journal of Materials Chemistry B. 2017;5(33):6786-93.”
- Kubera Sampath Kumar S, Prakash C, Subramanian S. Study on performance of different wound dressings on surgical non infected wounds. Journal of Natural Fibers. 2021;18(2):161-74.
- Yang Y, Hu H. Spacer fabric-based exuding wound dressing–Part II: Comparison with commercial wound dressings. Textile Research Journal. 2017;87(12):1481-93.
- Tong S-f, Yip J, Yick K-l, Yuen C-wM. Exploring use of warp-knitted spacer fabric as a substitute for the absorbent layer for advanced wound dressing. Textile Research Journal. 2015;85(12):1258-68.
Furthermore, to address the need of a practical clinical outcome we reported a preliminary clinical study on the evaluation of efficacy and tolerability of our triple-layer gauze for the treatment of chronic lesions in 20 patients. Please see the reference:
- “Pittarello, Monica and Elia Ricci. "Valutazione clinica dell’efficacia e della tollerabilità di un nuovo presidio a base di Rigenase® e poliesanide nel trattamento di lesioni cutanee croniche." (2022).”

Round 2
Reviewer 1 Report
The authors have provided sufficient responses to my comments, it can be accepted for publication